# Investigation of the Densification Behavior of Alumina during Spark Plasma Sintering

**DOI:** 10.3390/ma15062167

**Published:** 2022-03-15

**Authors:** Maksim S. Boldin, Alexander A. Popov, Evgeni A. Lantsev, Aleksey V. Nokhrin, Vladimir N. Chuvil’deev

**Affiliations:** Materials Science Department, Research and Development Institute of Physics and Technology, National Research Lobachevsky State University of Nizhny Novgorod, Gagarin Ave. 23/3, 603022 Nizhny Novgorod, Russia; popov@nifti.unn.ru (A.A.P.); elancev@nifti.unn.ru (E.A.L.); nokhrin@nifti.unn.ru (A.V.N.); chuvildeev@nifti.unn.ru (V.N.C.)

**Keywords:** alumina, spark plasma sintering, densification, sintering stages

## Abstract

The article presents the results of the investigation of the mechanism of the densification behavior of alumina-based ceramics during spark plasma sintering. The role of the heating rates and additives were investigated. The first (initial) stage of sintering was investigated by the Young–Cutler model. The second (intermediate) stage of sintering was investigated as a process of plastic deformation of a porous body under external pressure. It was shown that, at the initial stage, the formation of necks between the particles is controlled by grain boundary diffusion (the activation energy is *Q_b_* ≈ 20 kT_m_). At this stage, accommodation of the shape of the alumina particles is also occurring (an increase in the packing density). The accommodation process facilitates the shrinkage of the powder, which is reflected in a decrease in the effective activation energy of shrinkage at low heating rates (10 °C/min) to *Q_b_* ≈ 17 kT_m_. At heating rates exceeding 10 °C/min, the intensity of the processes of accommodation of alumina particles turns out to be much slower than the existing diffusion processes of growth of necks between the alumina particles. It was shown that the grain boundary sliding mechanism that occurs in the second stage of sintering can play a decisive role under conditions of spark plasma sintering with a high heating rate. The found value of the activation energy at the second stage of sintering is also close to the activation energy of grain–boundary diffusion of alumina (*Q_b_* ≈ 20 kT_m_). The influences of the second phase particles of MgO, TiO_2_, and ZrO_2_ on densification behavior of alumina-based ceramics were investigated. Since at the first stage of sintering the densification relates with the formation of necks between the particles of alumina, the additives (0.5% vol) have no noticeable effect on this process. It was also shown that the second phase particles which are located at the grain boundaries of alumina are not involved in the slip process during the second sintering stage. Analysis shows that additives act only in the final (third) stage of spark plasma sintering of alumina.

## 1. Introduction

The presented article is a detailed version of the publication in the collection of works International Conference on Synthesis and Consolidation of Powder Materials [1] and based on the PhD Thesis [2]. The combination of high hardness, high temperature, and chemical resistance makes alumina one of the promising ceramic materials [3,4]. The alumina-based ceramics are actively used in mechanical and power engineering [5,6]. Utilizing these properties of alumina to the full extent is only possible when ceramics have a fine-grained structure with a high relative density. The most common method of sintering alumina-based ceramics is conventional (pressureless) sintering or hot pressing [7,8] of green bodies. The mechanisms of sintering of alumina powders under these conditions have been well studied [9,10]. The disadvantage of this method is the intensive growth of grains, which often prevents the formation of a high-density structure with a small grain size [11,12]. In this regard, recently, methods of fast densification (shrinkage) of powders have become very popular. One of the more discussed methods in this series is Spark Plasma Sintering (SPS) [13,14,15,16,17]. This technology provides a high level of control of the sintering process under constant pressure and high heating rates (10–1000 °C/min) [18,19]. Many scholars point out a key advantage of SPS—its ability to produce the dense materials, including alumina, at low sintering temperatures [20,21]. In the process of solid-phase sintering of powder materials, it is customary to distinguish three stages. The first stage is densification of the powder up to ≈70% density—contacts are formed between the powder particles [10]. The second stage lies in the range from ≈70–90% density; at the stage of intensive densification, complex open porosity is formed. At the final stage from ≈90% to ≈100% density, the pores are healed and grains grow. From a practical point of view, of greatest interest during sintering under SPS conditions is the stage of growth of alumina grains in the process of spark plasma sintering and the influence of modifier additives (in the form of particles of the second phase) and physical and mechanical properties [22,23]. The introduction of the second phase particles is a common approach in the production of ceramic products in industry. The models are based on the concepts of the diffusion mechanism of mass transfer due to grain boundary and/or bulk diffusion and grain boundary migration (grain growth). Typically, sintering models consider a medium heating rate of powders with a grain size of up to 1 µm [10]. These models assume that the dominant mechanism of shrinkage in the final stage of sintering is a creep. However, the joint use of high heating rates and submicron powders can lead to a change in densification mechanisms. Thus, Ref. [24] reports about the viscous flow mechanism during spark plasma sintering of alumina but do not reveal its nature. In the range of temperatures and heating rates that are used in this study, the densification process can be described from the point of view of grain boundary sliding. It should be noted that a similar result was found for SPS nanopowders of zirconia [25]. There are few works with a detailed analysis of the sintering kinetics at the first and second stages of sintering and the study of the role of additives at each of these stages [26,27]. To describe the first stage of sintering, geometric models that promote shrinkage were used. Various particle shapes, vacancy sinks, and diffusion paths have been considered as they affect sintering shrinkage. These simplified models are extended to compacts of inhomogeneous particles, so that most of the sintering kinetics of a substance can be determined by measuring the shrinkage rate of powder compacts. To describe the second stage of sintering, models based on the diffusion resorption of pores and pore systems due to compacting sintering mechanisms (grain boundary, bulk diffusion) depending on the types of powders and sintering conditions have been developed.

In the present work, an attempt was made to analyze the sintering kinetics of pure alumina and alumina-based ceramics with the addition of particles of the second phase MgO, TiO_2_, and ZrO_2_ under the conditions of spark plasma sintering at the first stage within the framework of the Young–Cutler model. The second stage of sintering is described as a process of plastic deformation of a porous body under the influence of external pressure and temperature at various rates of deformation of the porous body. In the case of conventional sintering of alumina, this factor is not taken into account, but, under conditions of hot pressing with a high heating rate, it can play a decisive role.

The aim of this work is to analyze the densification behavior of fine alumina powder from a grain boundary sliding mechanism point of view. The authors also try to study the stages of spark plasma sintering of alumina powders, as well as to study the effect of particles of the second phase on the sintering kinetics at different stages of sintering.

## 2. Materials and Methods

The principal objects of our study are the submicron powders of alumina α-Al_2_O_3_, as well as powder compositions α-Al_2_O_3_ + 0.5 vol.% MgO, α-Al_2_O_3_ + 0.5 vol.%TiO_2_, and α-Al_2_O_3_ + 0.5 vol.% ZrO_2_. The additives were selected based on the different ways they interact with alumina: (i) ZrO_2_ is insoluble in Al_2_O_3_ and does not form compounds with it [28]; (ii) MgO is soluble in Al_2_O_3_ at low temperatures (T ~ 600 °C), at T > 600 °C, it forms the MgAl_2_O_4_ spinel phase [29]; (iii) TiO_2_ is soluble in Al_2_O_3_ at high temperatures (T > 1000 °C), at T > 1200 °C, it forms the Al_2_TiO_5_ phase [30]. The characteristics of the initial powders are shown in Table 1.

Powder compositions were obtained by mixing the initial components in a Fritsch Pulverisette 6 planetary mill (Idar-Oberstein, Germany)at a mixing jar rotation frequency of 200 rpm for 20 h. Mixing was carried out in methanol; grinding bodies and accessories were made of zirconia. The samples were sintered in graphite molds using the DR. SINTER model SPS-625 (SPS Syntex Inc., Tokyo, Japan). The temperature was measured from the outer (cylindrical) surface of the mold through a hole in the graphite felt and was controlled using a radiation pyrometer, which is part of the setup. The sintering kinetics was studied at a constant heating rate. The powders were heated to the sintering temperature T = 1520 °C at different heating rates. Heating was carried out in a pulse mode 12:2. At the end of sintering, the samples were cooled with the equipment turned off. Sintering was carried out in vacuum (6 Pa). Sintering pressure was 70 MPa.

The shrinkage of the samples during sintering was monitored using a precision dilatometer of the DR. SINTER model SPS-625 machine. The shrinkage measurement accuracy was ±0.05 mm. To eliminate the effect of the thermal expansion of the “machine-sample” system on the dilatometer measurement of the shrinkage, we additionally studied the thermal expansion of the system without samples (with an empty mold). The recalculation of the shrinkage into the relative density value $\rho /\rho_{th}$ was carried out according to the procedure described in [31] (here, $\rho_{th}$ is the theoretical density of the system). The correctness of the conversion was checked by comparing the calculated value of $\rho /\rho_{th}$ with the experimentally measured relative density of the sintered ceramic.

The density of sintered samples (ρ) was measured by hydrostatic weighing in distilled water using a Sartorius CPA scale. The measurement accuracy was ±0.005 g/cm^3^. The theoretical density of ceramics was taken equal to $\rho_{th}$ = 3.992 g/cm^3^ for α-Al_2_O_3_, $\rho_{th}$ = 3.989 g/cm^3^ for α-Al_2_O_3_ + 0.5% vol. MgO, $\rho_{th}$ = 3.993 g/cm^3^ for α-Al_2_O_3_ + 0.5% vol. TiO_2_, $\rho_{th}$= 4.0 g/cm^3^ for the composition α-Al_2_O_3_ + 0.5%vol. ZrO_2_. The theoretical values of the density were calculated based on the analysis of the results of X-ray studies. The error in the relative density $\rho /\rho_{th}$ was 0.2%. Vickers hardness ($H_{v}$) was determined using a Qness A60+ microhardness tester (ATM Qness GmbH, Salzburg, Austria) by measuring the lengths of the diagonals of the diamond pyramid (indenter) imprint on the polished surface of the sample under a load of 2 kg (19.6 N). The loading time was 30 s. The indenter was a diamond pyramid with an apex angle of 136° and a diagonal length of 500 μm. The fracture toughness ($K_{IC}$) was determined by measuring the crack lengths from the Vickers indenter indentation angles. The minimum $K_{IC}$ values were calculated using the Palmqvist method [32], based on the length of the longest crack. When calculating $K_{IC}$, the value of the elastic modulus was taken equal to E = 380 GPa. 

X-ray phase analysis (XRD) was carried out using a Shimadzu XRD-7000 X-ray diffractometer (Shimadzu Europa GmbH, Duisburg, Germany) with the following scanning parameters: copper radiation, scanning range of angles 20–90°; scanning step 0.04°; exposure duration at each point 3 s; sample rotation speed 60 rpm. The diffractograms were interpreted in DIFFRACplus Evaluation package Release 2009 and the PDF-2 database Release 2009. The microstructure of the samples was studied using a Jeol JSM-6490 scanning electron microscope (Jeol Ltd., Tokyo, Japan) and a Jeol JEM 2100 transmission electron microscope (Jeol Ltd., Tokyo, Japan). The average grain size in the sintered ceramics was calculated by the secant method based on the analysis of at least 200 grains. The error in measuring the average grain size was ±0.2 μm.

## 3. Results and Discussion

### 3.1. XRD Analysis of Alumina Powder

The XRD results (Figure 1) show that the alumina powder consists of 100% α-Al_2_O_3_ phase. The magnesium oxide powder consists of an MgO phase with a cubic structure. Titanium oxide powder contains two main phases—anatase (~74% by volume) and rutile (~26% by volume), which are various polymorphic modifications of titanium oxide. Zirconium oxide powder ZrO_2_-3%Y_2_O_3_ contains a monoclinic (~58% vol.) and tetragonal (~42% vol.) phase.

### 3.2. Analysis of the Densification Behavior

Figure 2 shows the dependence of the shrinkage on the heating temperature $\rho /\rho_{th}\left(T\right)$ for powders based on alumina. As seen from Figure 2, dependencies $\rho /\rho_{th}\left(T\right)$ have three stages. In Figure 2, the temperature dependences shrinkage at all investigated heating rates have three stages. Such type of densification cure is common for solid phase sintering of polycrystalline materials [10].

Graphs of the dependence of the shrinkage rate on the heating temperature of ceramic samples based on alumina are shown in Figure 3. It can be seen from the graphs that the maximum shrinkage rate ($S_{max}$) at one heating rate differs for different powder systems. The minimum $S_{max}$ values are observed for pure alumina. The maximum values of the shrinkage rate at low heating rates (10 °C/min) are observed for ceramics with the addition of TiO_2_, and at high heating rates (350–700 °C/min)—for ceramics with MgO and ZrO_2_ additives. An increase in the heating rate leads to an increase in *S_max_* for all powders. This result correlates with [33]. When using high heating rates, the isolated powder particles are placed under the action of the high temperatures; at this moment, the necks between the particles are formed and the saved surface curvature of the particles provides an additional driving force for sintering and $S_{max}$ increases.

Thus, for pure α-Al_2_O_3_, an increase in the heating rate from 10 to 700 °C/min increases $S_{max}$ from 2.7·10^−3^ to 1.2·10^−1^ m/s for α-Al_2_O_3_ + 0.5%vol. The TiO_2_ heating rate increase from 10 to 700 °C/min increases the maximum sintering rate from 7.0·10^−3^ to 1.1·10^−1^ m/s.

### 3.3. Study of the Structure and Mechanical Properties of Ceramics

The mechanical properties of ceramics based on α-Al_2_O_3_ are shown in Table 2. Figure 4 shows the dependence of the properties of ceramics on the heating rate. Microstructure of ceramics obtained by scanning electron microscopy are shown in Appendix A (Figure A1, Figure A2, Figure A3 and Figure A4). 

An increase in the heating rate (V) from 10 °C/min to 700 °C/min leads to a monotonous decrease in the relative density of sintered samples (Figure 4a). For ceramics obtained from pure α-Al_2_O_3_ powder and α-v + 0.5% vol. MgO, reduction in the value of $\rho /\rho_{th}$ is 0.2–0.4%; for compositions α-Al_2_O_3_ + 0.5% vol. TiO_2_ and α-Al_2_O_3_ + 0.5% vol. ZrO_2_ decrease in relative density, $\rho /\rho_{th}$ is 0.7–1.1%. The decrease in the relative density of ceramics with an increase in the sintering rate is associated with residual micro and macropores that did not have time to heal at high heating rates, and the second phase particles inhibit neck growing at the initial sintering stage and suppress the densification at the following sintering stages [34].

An increase in the heating rate leads to a decrease in the average grain size of the sintered ceramic (Figure 4b). For ceramics obtained from pure α-Al_2_O_3_ powder, an increase in the heating rate from 10 °C/min to 350 °C/min (at Т = 1520 °C) leads to a sharp decrease in the average grain size from d = 5.1 μm to d = 1.9 μm, while further increase in the heating rate to V = 700 °C/min does not affect the size of the grain. This is because the scale of grain change with increasing heating rate from 350 °C/min to 700 °C/min is less than the measurement error. A similar dependence is observed for ceramics α-Al_2_O_3_ + 0.5% vol. TiO_2_: an increase in the heating rate from 10 °C/min to 350 °C/min leads to a decrease in the average grain size from d = 7.2 µm to d = 2.4 µm. For ceramics α-Al_2_O_3_ + 0.5% vol. MgO and α-Al_2_O_3_ + 0.5%vol. ZrO_2_, the decrease in the average grain size with an increase in the heating rate is much less pronounced than for pure α-Al_2_O_3_, and throughout almost the entire range of heating rates, the average grain size in the sintered ceramic is approximately 1–1.5 μm. Moreover, the average grain size in alumina with the addition of 0.5 vol.%TiO_2_ turns out to be greater than in pure alumina sintered under the same temperature and heating rate, while the average grain size in ceramics with an addition of 0.5 vol.% MgO and 0.5 vol.% ZrO_2_ is less than in pure alumina.

Figure A1, Figure A2, Figure A3 and Figure A4 show that, with increasing sintering speed, the grain structure becomes fine-grained and more uniform. With an increase in the sintering rate from 10 to 700 °C/min, the residence time of the powder at a high temperature is reduced and the grains do not have time to grow.

X-ray diffraction analysis of a sample of α-Al_2_O_3_ + 0.5% vol. MgO, shown in Figure 5, shows the presence of the second phase in the structure of particles—MgAl_2_O_4_ spinel. However, as shown in Figure A2, in the phase contrast mode does not show any particles of the second phase. This is because the atomic numbers of aluminum and magnesium are very close. In the structure of samples of the system α-Al_2_O_3_ + 0.5% vol. ZrO_2_ (Figure A3) and α-Al_2_O_3_ + 0.5% vol., TiO_2_ (Figure A4) particles of the second phase are observed.

The dependences of the ceramic microhardness on the heating rate correlate with the dependences of the grain size on the heating rate (Figure A1c). A decrease in the grain size with an increase in the heating rate from 10 °C/min to 350 °C/min leads to an increase in microhardness by an average of 7%. A further increase in the rate does not have a significant effect on the size of the grain, and, consequently, on the microhardness. The highest microhardness is reached by ceramics produced from the composition α-Al_2_O_3_ + 0.5% vol. MgO ($H_{v}$ = 19.8–21.4 Gpa), the lowest—by ceramics produced from the composition α-Al_2_O_3_ + 0.5% vol. TiO_2_ ($H_{v}$ = 16.0–17.1 Gpa). At higher heating rates (350–700 °C/min), a slight decrease in the hardness of ceramics is observed, which is probably due to the influence of residual porosity on the mechanical properties (strength) of the ceramics. An increase in the heating rate has no significant effect on the fracture toughness of ceramics (Figure A1d). This can be explained by the fact that a decrease in the average grain size with an increase in the heating rate is compensated by an increase in residual porosity [10]. Fracture toughness of ceramics obtained from pure α-Al_2_O_3_ powder, as well as compositions α-Al_2_O_3_ + 0.5% vol. MgO, α-Al_2_O_3_ + 0.5% vol. ZrO_2_, is $K_{IC}$ ~ 2.5 Mpa·m^1/2^. Fracture toughness of ceramics α-Al_2_O_3_ + 0.5% vol. TiO_2_ is $K_{IC}$ ~ 3 Mpa·m^1/2^.

### 3.4. Analysis of Shrinkage Curves

The form of the $\rho /\rho_{th}\;\left(T\right)$ dependence shown in Figure 3 is typical for solid-phase sintering [10]. This suggests that the processes occurring during SPS of alumina-based ceramics can be described as a sequence of processes of the initial (I), intermediate (II), and final (III) stages of sintering [10]. According to [10], the initial stage of sintering (stage I) is characterized by the formation of contacts between powder particles. At a heating rate V = 10 °C/min, this phase is observed at temperatures up to 1125 °C (Figure 6a). The transition between stages I and II occurs at $\rho /\rho_{th}$ ~ 0.7 [10]. Stage II (~1125–1230 °C at V = 100 °C/min) is characterized by an increase in the contact area between the particles and intense shrinkage of the powder. As shown above, the temperature of the beginning of the stage of intensive shrinkage of alumina powders does not depend on the presence of MgO, ZrO_2_, and TiO_2_ additives at a concentration of 0.5% vol. It is important to emphasize that at stage II, the growth of grains in ceramics is almost absent. Therefore, for example, for pure Al_2_O_3_ obtained by SPS at a temperature T = 1400 °C corresponding to the temperature of the end of stage II at a heating rate of 700 °C/min (Figure 6b), the relative density is $\rho /\rho_{th}$ = 0.92, and the average grain size (d ~ 0.2–0.3 μm) is close to the initial particle size of the original powder (Figure A1a). After the end of stage II, the structure of the ceramic contains pores ranging in size from 50 to 150 nm (Figure 6b) located along the grain boundaries of alumina.

Note also that, during sintering, the grains retain their equiaxed shape (Figure 7), while, under ordinary plastic deformation, the alumina grains change their shape and stretch in the direction of deformation of the sample [35,36].

At stage III (at T > 1230 °C at V = 100 °C/min, $\rho /\rho_{th}$ ~ 0.9), closed porosity is formed, and diffusion-controlled dissolution of isolated pores and grain growth occurs. As an example, Figure 7 shows images of the polished surface of an Al_2_O_3_ sample obtained at T = 1520 °C (V = 100 °C/min). The relative density of the sample is, $\rho /\rho_{th}$ = 0.99, the average grain size is d ~ 3.0 μm. As can be seen from Figure 7a, after the end of stage III, large pores are not observed. Single pores can be observed at the grain boundaries, but their size does not exceed 50 nm (Figure 7b).

### 3.5. First Stage Analysis

Let us analyze the mechanism of shrinkage at the initial stage of sintering (Stage I). To analyze the kinetics of powder sintering at the initial stages of sintering, we will use the model presented in [26,37]. The model describes the initial stage of non-isothermal sintering of spherical particles during simultaneous volumetric and grain–boundary diffusion:(1)$$\varepsilon^{2}\frac{\partial \varepsilon }{\partial t}≃A_{1}\left(\frac{G\Omega }{kT}\right)\left(\frac{\gamma D_{v}}{Gd^{3}}\right)\varepsilon +A_{2}\left(\frac{G\Omega }{kT}\right)\left(\frac{\gamma \delta D_{b}}{Gd^{4}}\right)$$
where ε—relative shrinkage, t—sintering time ($t=T_{sint}/V$, V—heating rate, $T_{sint}$—sintering temperature), γ—surface free energy, $D_{v}$—volume diffusion coefficient, $D_{b}$—grain boundary diffusion coefficient, δ—grain boundary width, d—grain size, k—Boltzmann constant, $A_{1}$ and $A_{2}$—constants ($A_{1}$ = 0.32, $A_{2}$ = 0.04). The grain boundary $D_{b}$ and bulk $D_{v}$ diffusion coefficients correspond to the Arrhenius form $D_{b}=D_{b0}exp\left(-Q_{b}/RT\right)$ and $\;D_{v}=D_{v0}exp\left(-Q_{v}/RT\right)$, where $D_{b0}$ and $D_{v0}$ are pre-exponential coefficients, $Q_{b}$ and $Q_{v}$ are the activation energies of grain boundary and volume diffusion accordingly, R—gas constant.

In the case of dominance of bulk diffusion, Equation (1) after integration over T reduces to the expression [37]:(2)∂ε∂T≃2A1γDv0RT2Gd3VQvGΩkT 12Qv2RT2 exp−Qv2RT

In the case of dominance of grain boundary diffusion, Equation (1) after integration over T and transformations reduces to the expression [37]:(3)∂ε∂T≃3A2 γDb0RT2Gd3VQbGΩkT13Qb3RT2 exp−Qb3RT

In accordance with [26], the slope of the temperature dependence of shrinkage in coordinates $\ln\left(T\;\partial \varepsilon /\partial T\right)-T_{m}/T$ corresponds to the effective activation energy of sintering $mQ_{tg}$ ($Q_{tg}-$ slope tangent), where *m* is a coefficient depending on the dominant sintering mechanism (m = 1/3—for grain boundary diffusion (Equation (2)), m = 1/2—for bulk diffusion (Equation (3)), $T_{m}$ = 2323 K is the melting point of alumina. When calculating the activation energy, the melting point of alumina was assumed to be independent from the presence of additives. Figure 8 shows the $\ln\left(T\;\partial \varepsilon /\partial T\right)-T_{m}/T$ for heating rates of 10, 100, 350 and 700 °C/min, where the slope tangent corresponds to $mQ_{tg}$. Physically correct value of activation energy ($Q=mQ_{tg}$) is obtained at m = 1/3 (16.8 ÷ 19.9 kT_m_) than at m = ½ (11.2 ÷ 13.3 kT_m_). Table 3 shows the values of the effective activation energy of sintering (Q) at the first stage of sintering at heating rates of 10–700 °C/min at m = 3. The dependence of the effective activation energy of sintering for systems based on alumina on the heating rate is shown in Figure 9.

The effective activation energy of shrinkage for pure alumina with an increase in the heating rate from 10 to 700 °C/min increases from 16.9 kT_m_ (at 10 °C/min) to 19.6 kT_m_ at a rate of 100–700 °C/min. The calculated value of the activation energy of shrinkage for pure alumina is comparable to the energy of grain boundary diffusion in alumina ($Q_{b}$ ≈ 20 kT_m_ [38]).

The analysis shows that the introduction of additives does not lead to a change in the effective activation energy of shrinkage at the initial stage of sintering. At the initial stage of sintering, high contact stresses arise in the contact area of alumina particles [39], which ensures the sliding of particles relative to each other, thereby leading to their accommodation (an increase in the packing density of particles). The accommodation process facilitates the shrinkage of the powder, which is reflected in a decrease in the effective activation energy of shrinkage at low heating rates (10 °C/min). At heating rates exceeding 10 °C/min, the intensity of the processes of accommodation of alumina particles turns out to be much slower than the existing diffusion processes of growth of necks between the alumina particles. 

The densification mechanism was established by comparing the obtained values of the effective activation energy with the tabular activation energies of the processes of grain boundary and volumetric diffusion. It is shown that only at m = 1/3 does the value of the effective activation energy correspond to the value characteristic of alumina. Thus, in our opinion, the mechanism providing powder shrinkage at the first stage of sintering under conditions of high-speed heating is grain–boundary diffusion. Grain–boundary diffusion ensures the growth of necks between particles of alumina. Under such conditions, the additives have no noticeable effect on the shrinkage of alumina.

### 3.6. Second Stage Analysis

According to [35,36], the process of intensive shrinkage of a powder material can be described as the process of plastic deformation of a solid porous material. In this case, the maximum strain rate (${\dot{\varepsilon }}_{max}$) can be determined experimentally (Table 4) as:(4)$${\dot{\varepsilon }}_{max}=\frac{S_{max}\;}{L_{initial}}$$
where $L_{initial}$ ~ initial height of the sintered sample, $S_{max}$—maximum shrinkage rate.

Let us assume that the process of shrinkage of a submicron powder material can be described as a process of plastic deformation [35,36]:(5)$${\dot{\varepsilon }}_{max}=A{\left(\Phi \frac{\sigma }{G}\right)}^{2}\left(\frac{G\Omega }{kT}\right){\left(\frac{b}{d}\right)}^{2}\frac{\delta D_{b0}}{b^{3}}$$
where A = 3.34—constant [38], G = 126 GPa—shear modulus of alumina [38], σ = 70 Mpa—stress at which sintering occurs, Ω = 4.26·10–29 m^3^—atomic volume [38], δ=2b—width of the boundary grains, b = 4.76·10–10 m is the Burgers vector [38], $\Phi =1/\left(1-f_{pore}\right)$—coefficient that takes into account the effect of porosity on stresses, $f_{pore}$—volume fraction of pores (can be calculated as a first approximation based on the data on the relative density of the material $f_{pore}=1-\rho /\rho_{th}\;$).

The expression for the grain boundary diffusion coefficient $D_{b}$ has the form:(6)$$D_{b}=D_{b0}exp\left(-\frac{Q_{b}}{kT}\right)$$

If we equate (4) and (5) and substitute constants given above and also take $\delta D_{b0}$ = 10^−8^ (m^3^/s) [40] (the case of grain–boundary diffusion of oxygen ions) at a temperature $T_{max}$ corresponding to the maximum shrinkage rate $S_{max}=S\left(T_{max}\right)$, we can calculate the value of the diffusion activation energy at the second stage of sintering. The calculated values are also presented in Table 4. 

Table 4 shows that the found values of the activation energy at the second stage of sintering are close to the activation energy of grain–boundary diffusion of alumina and amount to $Q_{b}$ ~ 20 kT_m_ (or 380 kJ/mol) [38]. The high value of the deformation rate (~10^−3^ s^−1^) and the value of the activation energy of the deformation process, corresponding to the activation energy of grain–boundary diffusion, indirectly indicate that intense shrinkage at the second stage of sintering unfolds according to the mechanism of grain–boundary sliding. Note that the activation energy of the shrinkage process at the second stage depends weakly on the presence of MgO, ZrO_2_, and TiO_2_ additives. Thus, the particles of the second phase do not affect the rate of grain boundary sliding and the kinetics of shrinkage of alumina. Figure 10 shows a diagram explaining the reason for the lack of influence of MgO, ZrO_2_ and TiO_2_ additives on the densification behavior of Al_2_O_3_ at the second stage of sintering. Shrinkage of the powder system under the action of an applied load is caused by deformation and grain boundary sliding of Al_2_O_3_ grains; particles of the second phase located at the boundaries of Al_2_O_3_ grains are not involved in the slip process.

The expression describing the rate of plastic deformation during grain boundary sliding controlled by second-phase particles lying at the grain boundaries of the base material has the following form [36]:(7)$${\dot{\varepsilon }}_{p}=A_{3}\left(\Phi \frac{\sigma }{G}\right)\left(\frac{\delta D_{b}}{b^{3}}\right){\left(\frac{b}{r_{a}}\right)}^{3}$$
where ${\dot{\varepsilon }}_{p}$—deformation rate controlled by grain boundary sliding in a material containing particles of the second phase, $A_{3}$ ~ 1—constant, $r_{a}$—size of particles of the second phase.

With the particle size of the second phase $r_{a}$ ~ 10–100 nm, σ/G ~ 0.0005.5 and Φ ~ 1.25, we obtain ${\dot{\varepsilon }}_{p}$/$\dot{\varepsilon }$ ~ 100. If grain boundary sliding had been controlled by particles of the second phase of such a small size, the plastic deformation rate would have exceeded the plastic deformation rate calculated by another method by two orders of magnitude. Thus, the rate of deformation controlled by grain boundary sliding of Al_2_O_3_ grains at the second stage of sintering cannot be limited by MgO, TiO_2_ и ZrO_2_ particles due to their small size.

It should also be noted that grain growth is absent at the analyzed (second) sintering stage; therefore, the traditional approach for determining the mechanism of plastic deformation based on plotting the dependence of the deformation rate on the grain size cannot be used.

The results obtained correlate with the results of [24] where the viscous flow was called as a major mechanism of spark plasma sintering of alumina, but authors [24] do not reveal its nature. Viscous flow should have led to obtaining the elongated grains which were not observed. Another kind of such “plastic deformation” without elongation of grains can be caused by grain boundary sliding during sintering. It should be noted that a similar result was found for SPS nanopowders of zirconia [25].

Since grain growth occurs at the third stage of sintering, the densification behavior at this stage will depend on the grain growth kinetics (the ratio of the grain boundary migration rate and the pore migration rate) [10]. For this reason, the analysis of the third stage of sintering is excluded from consideration in this article.

## 4. Conclusions

Analysis of the densification behavior of fine alumina powder at different stages of spark plasma sintering has been made. The first (initial) stage of sintering was investigated by the Young–Cutler model. It was shown that, at the initial stage, the formation of necks between the particles is controlled by grain boundary diffusion (the activation energy is *Q_b_* ≈ 20 kT_m_). It was shown that the joint use of high heating rates and submicron powders can lead to a change in densification mechanisms at the second (intermediate) stage of sintering. For that reason, the second stage of sintering was investigated from a grain boundary sliding mechanism point of view. From the above results, it is suggested that the plastic deformation of a porous body is also controlled by grain boundary diffusion (the activation energy is *Q_b_* ≈ 20 kT_m_).

The role of the heating rates of spark plasma sintering was investigated. It is shown that, at heating rates in the range from 10 °C/min to 700 °C/min, the transition temperature from the initial stage to the stage of intense shrinkage is Т = 1150 °C and does not depend on the heating rate. At low heating rates (10 °C/min), the effective activation energy of sintering at the initial stage is *Q_b_* ≈ 17 kT_m_, but already becomes equal to *Q_b_* ≈ 20 kT_m_ at rates above 100 °C/min. It can be caused by the accommodation process of an initial alumina powder particles, which is reflected in a decrease in the effective activation energy of shrinkage at low heating rates. It was shown that the heating rate does not affect the effective activation energy of sintering at the intermediate stage.

The influences of the second phase particles of MgO, TiO_2_, and ZrO_2_ on densification behavior of alumina-based ceramics were investigated. Since at the first stage of sintering the densification relates with the formation of necks between the particles of alumina, the additives (0.5% vol) have no noticeable effect on this process. It was also shown that the second phase particles which are located at the grain boundaries of alumina are not involved in the grain boundary sliding process during the second sintering stage. Analysis shows that additives act only at the final (third) stage of spark plasma sintering of alumina. It has been established that the introduction of MgO and ZrO_2_ additives inhibits grain growth, while TiO_2_ accelerates the growth. As a result, the hardness of ceramics with a fine grain size increases and falls with a large one. The crack resistance of ceramics increases with the addition of TiO_2_, which may be due to the intergranular crack movement along the “islands” of the aluminum titanite spinel [41].

## Figures and Tables

**Figure 1 materials-15-02167-f001:**
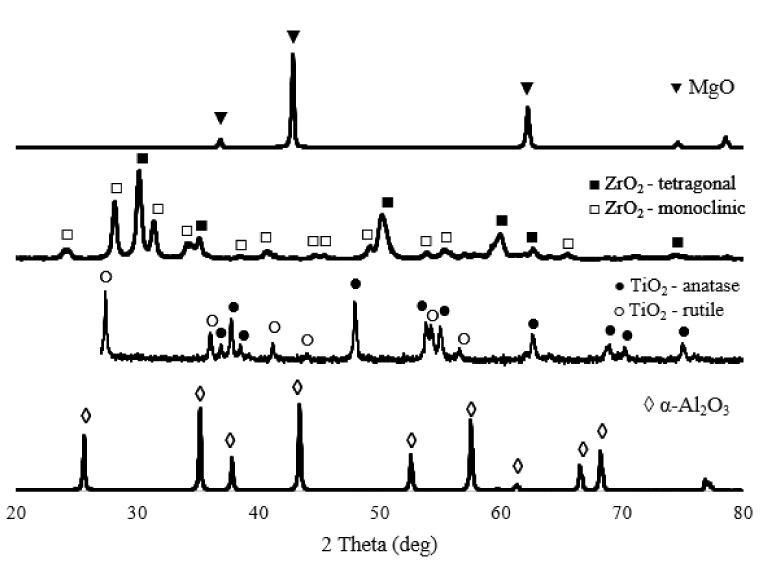
XRD patterns of powders.

**Figure 2 materials-15-02167-f002:**
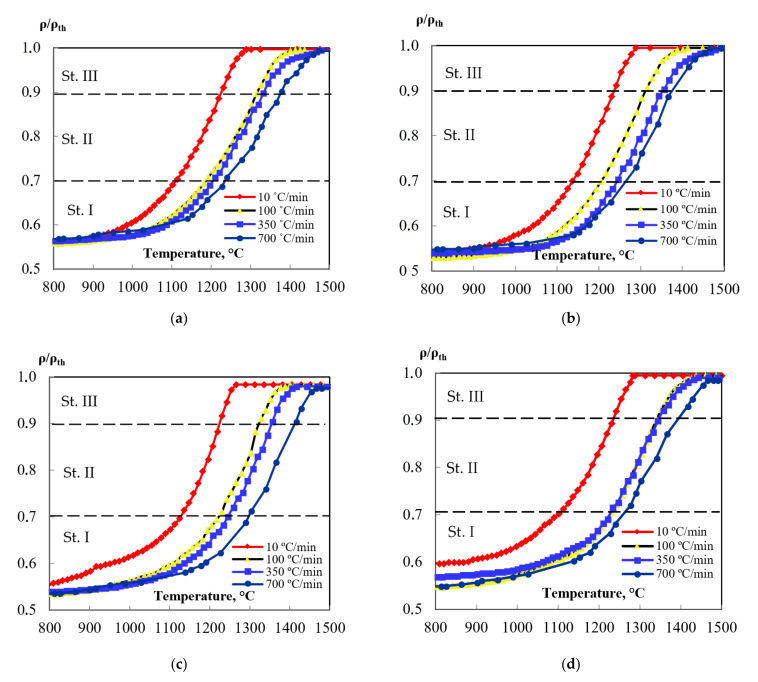
Shrinkage behaviors $\rho /\rho_{th}\left(T\right)$ for: (**a**) α-Al_2_O_3_, (**b**) Al_2_O_3_ + 0.5% vol. MgO, (**c**) Al_2_O_3_ + 0.5%vol. TiO_2_, (**d**) Al_2_O_3_ + 0.5%vol. ZrO_2_—from the PhD thesis [2].

**Figure 3 materials-15-02167-f003:**
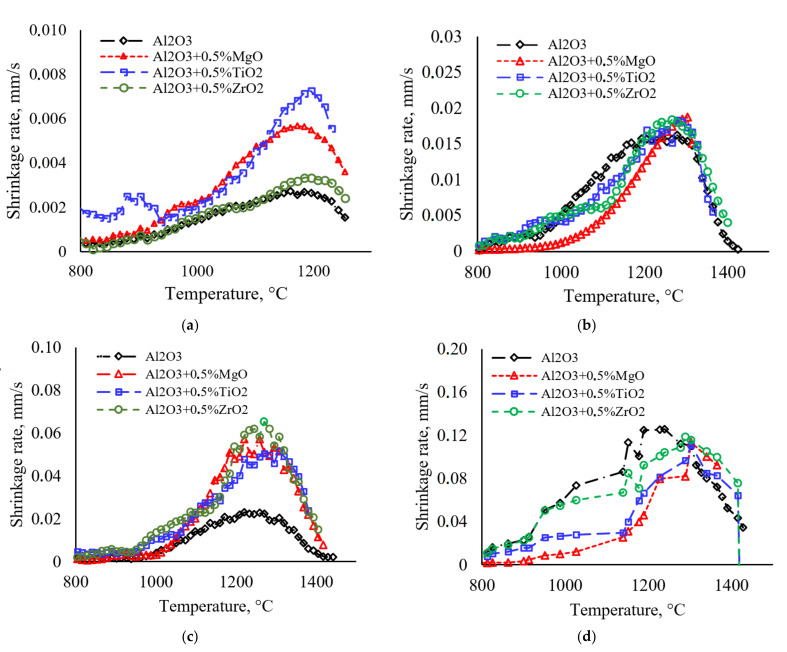
Shrinkage behaviors $S\left(T\right)$ at: (**a**) 10 °C/min, (**b**) 100 °C/min, (**c**) 350 °C/min, (**d**) 700 °C/min.

**Figure 4 materials-15-02167-f004:**
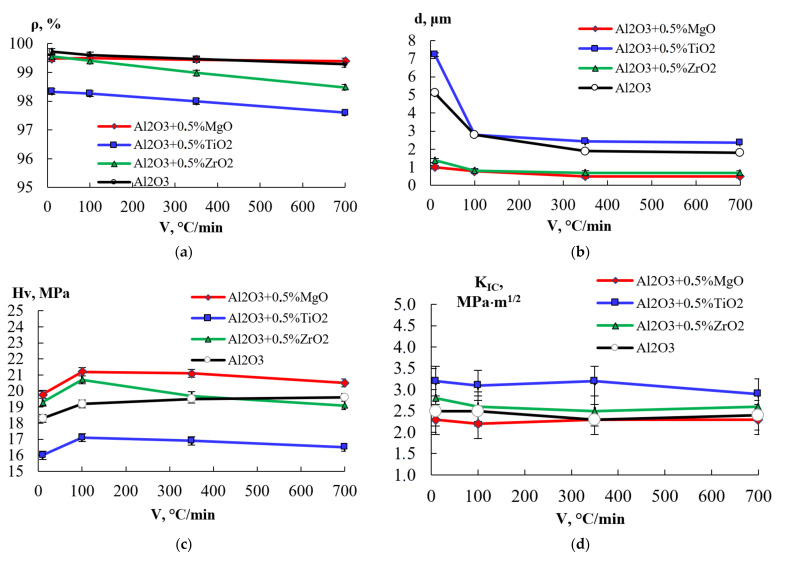
Mechanical properties: (**a**) density $\rho \left(V\right)$, (**b**) grain size $d\left(V\right)$, (**c**) hardness $H_{v},\left(V\right)$, (**d**) fracture toughness $K_{IC}\left(V\right)$.

**Figure 5 materials-15-02167-f005:**
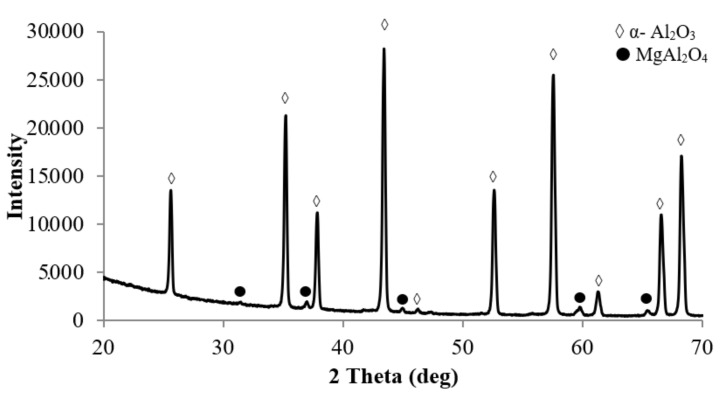
XRD patterns of the α-Al_2_O_3_ + 0.5%vol. MgO sample.

**Figure 6 materials-15-02167-f006:**
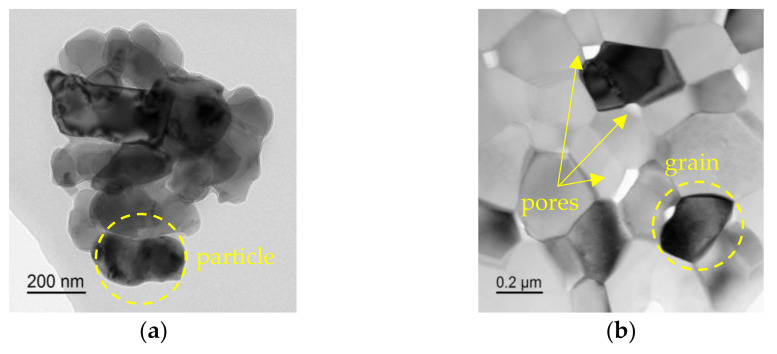
TEM images of Al_2_O_3_: (**a**) initial powder, (**b**) sample obtained at Т = 1400 °C, V = 700 °C/min ($\rho /\rho_{th}$ = 0.92, d ~ 0.2–0.3 μm). The pores are marked with arrows.

**Figure 7 materials-15-02167-f007:**
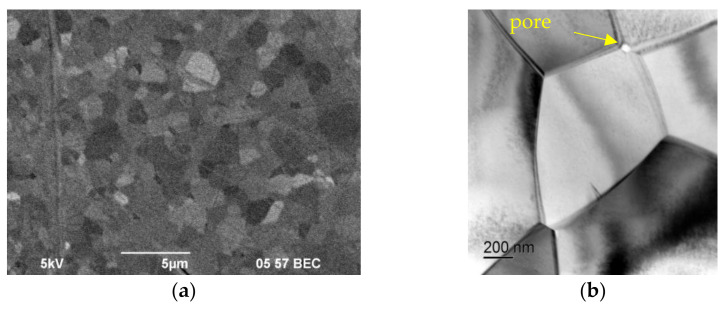
Microstructure of Al_2_O_3_ sample (Т = 1520 °C, V = 100 °C/min): (**a**) SEM, (**b**) TEM. The pores are marked with arrows.

**Figure 8 materials-15-02167-f008:**
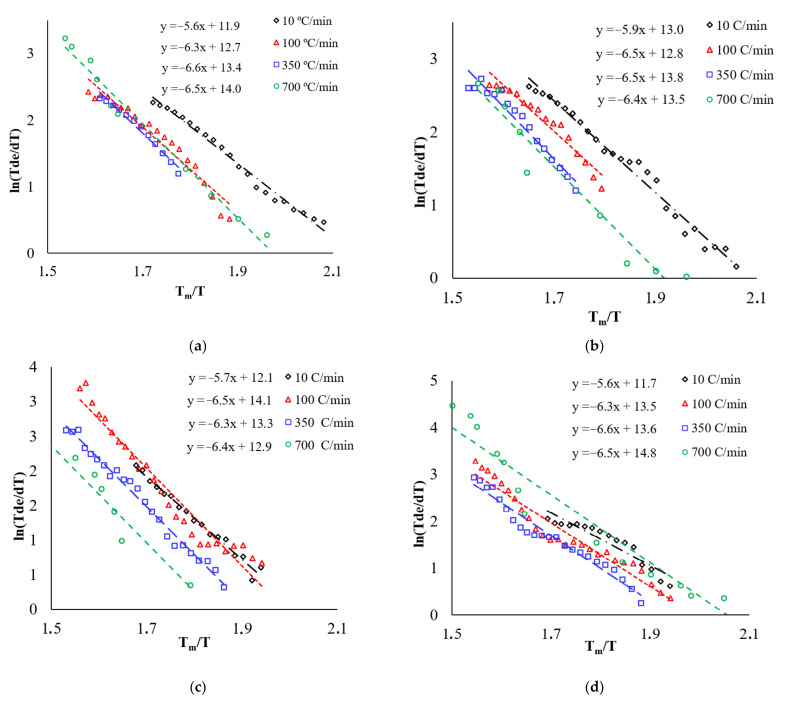
The dependence of $\ln\left(T\;\partial \varepsilon /\partial T\right)-T_{m}/T$ for: (**a**) Al_2_O_3_, (**b**) Al_2_O_3_ + 0.5%vol. MgO, (**c**) Al_2_O_3_ + 0.5%vol. TiO_2_, (**d**) Al_2_O_3_+ 0.5%vol. ZrO_2_.

**Figure 9 materials-15-02167-f009:**
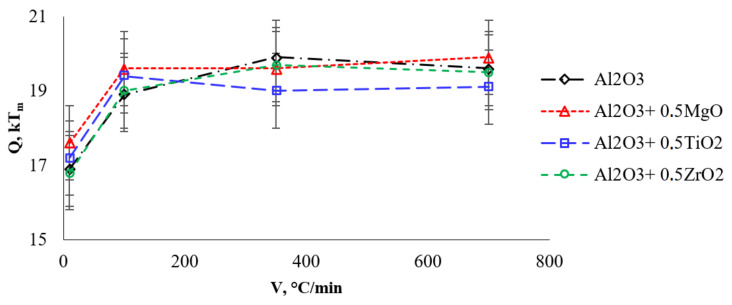
The dependence of Q—V on the first stage.

**Figure 10 materials-15-02167-f010:**
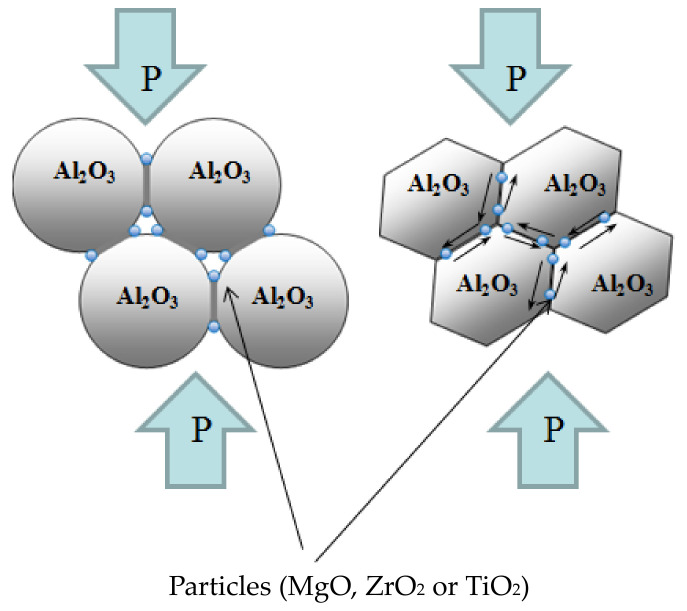
Scheme of arrangement of particles of the second phase.

**Table 1 materials-15-02167-t001:** Description of the starting powders.

Material	Manufacturer	Phase Composition *	Particle Size *
α-Al_2_O_3_	Taimei Chemicals Co., Ltd. (Tokyo, Japan)	α-Al_2_O_3_ (rhombohedral) ~ 100%	~0.2 μm
MgO	Alfa Aesar—A Johnson Matthey Company (Kandel, Germany)	MgO (cubic) ~ 100%	~0.1 μm
TiO_2_	Institute of Electrophysics, Ural Branch of RAS (Yekaterinburg, Russia)	TiO_2_ (anatase) ~ 74%, TiO_2_ (rutile) ~ 26%	~0.5 μm
ZrO_2_ (3% mol. Y_2_O_3_)	Pangea Int., Ltd. (Shanghai, China)	ZrO_2_ (monoclinic) ~ 58%, ZrO_2_ (tetragonal) ~ 42%	<0.03 μm

* Based on research (see paragraph 3.1)

**Table 2 materials-15-02167-t002:** Mechanical properties.

System	V, °C/min	*ρ*, % (Δ ± 0.2)	*d*, μm (Δ ± 0.2)	$H_{v}$, GPa (Δ ± 0.5)	$K_{IC}$, MPa·m^1/2^ (Δ ± 0.1)
α-Al_2_O_3_	10	99.7	5.1	18.3	2.5
α-Al_2_O_3_ + 0.5%vol. MgO		99.5	1.0	19.8	2.3
α-Al_2_O_3_ + 0.5%vol. TiO_2_		99.3	7.2	16.0	3.2
α-Al_2_O_3_ + 0.5%vol. ZrO_2_		99.5	1.4	19.3	2.8
α-Al_2_O_3_	100	99.6	2.8	19.2	2.5
α-Al_2_O_3_ + 0.5%vol. MgO		99.5	0.8	21.2	2.2
α-Al_2_O_3_ + 0.5%vol. TiO_2_		98.3	2.8	17.1	3.1
α-Al_2_O_3_ + 0.5%vol. ZrO_2_		99.4	0.8	20.7	2.6
α-Al_2_O_3_	350	99.5	1.9	19.5	2.3
α-Al_2_O_3_ + 0.5%vol. MgO		99.4	0.5	21.1	2.3
α-Al_2_O_3_ + 0.5%vol. TiO_2_		98.0	2.4	16.9	3.2
α-Al_2_O_3_ + 0.5%vol. ZrO_2_		98.9	0.7	19.7	2.5
α-Al_2_O_3_	700	99.3	1.8	19.6	2.4
α-Al_2_O_3_ + 0.5%vol. MgO		99.4	0.5	20.5	2.3
α-Al_2_O_3_ + 0.5%vol. TiO_2_		97.6	2.4	16.5	2.9
α-Al_2_O_3_ + 0.5%vol. ZrO_2_		98.5	0.7	19.1	2.6

**Table 3 materials-15-02167-t003:** Effective activation energy of sintering Q, kT_m_ on the first stage of sintering (the accuracy of the determination of Q is 2 kT_m_).

System	Heating Rate, °C/min			
	10	100	350	700
Al_2_O_3_	16.9	18.9	19.9	19.6
Al_2_O_3_ + 0.5%vol. MgO	17.6	19.6	19.6	19.9
Al_2_O_3_ + 0.5%vol. TiO_2_	17.2	19.4	19.0	19.1
Al_2_O_3_ + 0.5%vol. ZrO_2_	16.8	19.0	19.7	19.5

**Table 4 materials-15-02167-t004:** Calculation of the activation energy of grain boundary sliding on the second stage of sintering.

System	*V*, °C/min	$S_{max},\;m/s$	${\dot{\varepsilon }}_{max},\;s^{-1}$	Q, $kT_{m}\;(\Delta \;\pm \;2)$
α-Al_2_O_3_	10	2.7·10^−3^	7.0·10^−4^	19.3
α-Al_2_O_3_ + 0.5% vol. MgO		5.7·10^−3^	4.6·10^−4^	19.0
α-Al_2_O_3_ + 0.5% vol. TiO_2_		7.0·10^−3^	7.2·10^−4^	19.9
α-Al_2_O_3_ + 0.5% vol. ZrO_2_		2.8·10^−3^	2.8·10^−4^	20.1
α-Al_2_O_3_	100	1.4·10^−2^	1.2·10^−3^	19.9
α-Al_2_O_3_ + 0.5% vol. MgO		1.8·10^−2^	1.5·10^−3^	19.3
α-Al_2_O_3_ + 0.5% vol. TiO_2_		1.6·10^−2^	1.6·10^−3^	19.7
α-Al_2_O_3_ + 0.5% vol. ZrO_2_		1.7·10^−2^	1.5·10^−3^	19.6
α-Al_2_O_3_	350	2.3·10^−1^	1.9·10^−3^	19.8
α-Al_2_O_3_ + 0.5% vol. MgO		5.7·10^−1^	4.7·10^−3^	19.0
α-Al_2_O_3_ + 0.5% vol. TiO_2_		5.1·10^−1^	5.2·10^−3^	19.0
α-Al_2_O_3_ + 0.5% vol. ZrO_2_		6.5·10^−1^	5.6·10^−3^	19.6
α-Al_2_O_3_	700	1.2·10^−1^	1.0·10^−3^	19.0
α-Al_2_O_3_ + 0.5% vol. MgO		1.2·10^−1^	9.2·10^−3^	19.9
α-Al_2_O_3_ + 0.5% vol. TiO_2_		1.1·10^−1^	1.1·10^−3^	20.3
α-Al_2_O_3_ + 0.5% vol. ZrO_2_		1.2·10^−1^	9.9·10^−3^	19.7

## Data Availability

Data is contained within the article.

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
