# Peer review of "Investigation of the Densification Behavior of Alumina during Spark Plasma Sintering"

_materials, 2022, doi:10.3390/ma15062167_

Round 1

Reviewer 1 Report

Mechanical properties of alumina-zirconia material are missing in table 2.

Fracture surfaces in figures 6-9 may be better comparable, if they will be grouped by the heating rate, not the material. Also not all images are shown in same magnification.

Figures 11 and 12 are missing, which significantly complicates review of the discussion.

In table 3, how the heating rates were chosen? In the text, it is written 10-700°C/min. Why are there two different values for 10°C/min?

Third stage analysis is missing from the discussion. Explanation of omitting is missing as well.

Reason for crack resistance improvement by TiO2 addition is mentioned only briefly.

Author Response

Response to the Reviewer's comments

Dear Reviewer!

Thank you for your thorough analysis of our article. We agree with most of your comments and, in accordance with them, we have conducted a revision of our article. Below are our responses to the Reviewer's comments.

Reviewer #1

  1. Mechanical properties of alumina-zirconia material are missing in table 2.

Authors: We agree with this comment. The properties of alumina-zirconia material has been added.

  1. Fracture surfaces in figures 6-9 may be better comparable, if they will be grouped by the heating rate, not the material. Also not all images are shown in same magnification.

Authors: We do not agree with this comment. To see the features of the structure of the material obtained at different heating rates, different magnifications are needed. Please note that we have moved the drawings to Appendix A

  1. Figures 11 and 12 are missing, which significantly complicates review of the discussion.

Authors: We agree with this comment. Figures 11 and 12 have been added.

  1. In table 3, how the heating rates were chosen? In the text, it is written 10-700°C/min. Why are there two different values for 10°C/min?

Authors: We agree with this comment. This is a mistake. It has been fixed.

  1. Third stage analysis is missing from the discussion. Explanation of omitting is missing as well.

Authors: We agree with this comment. The following comment has been added.

Since grain growth occurs at the third stage of sintering, the densification behavior at this stage will depend on the grain growth kinetics (the ratio of the grain boundary migration rate and the pore migration rate). For this reason, the analysis of the third stage of sintering is excluded from consideration in this article.

  1. Reason for crack resistance improvement by TiO2 addition is mentioned only briefly.

Authors: We agree with this comment. At the moment, the authors do not have an answer to the question why the crack resistance improves by TiO2 addition. Probably, titanium ions can increase the strength of aluminum oxide grain boundaries in the same way as yttrium does [DOI: 10.1126/science.1119839]. Since this explanation requires additional verification and generally does not affect the conclusions made in the article, we did not include it in the article.

The revisions in the paper are highlighted in yellow.

Best regards,

Maksim S. BOLDIN (Corresponding author)

PhD, Researcher,

Lobachevsky State University of Nizhny Novgorod

[email protected]

23 Gagarina ave., Nizhniy Novgorod, 603950, Russian Federation

Author Response

Response to the Reviewer's comments

Dear Reviewer!

Thank you for your thorough analysis of our article. We agree with most of your comments and, in accordance with them, we have conducted a revision of our article. Below are our responses to the Reviewer's comments.

Reviewer #2

The authors present a work entitled "Investigation of the densification behavior of alumina during spark plasma sintering". They studied the stages of spark plasma sintering of alumina powders, as well as the effect of particles of the second phase on the sintering kinetics at different stages of sintering. This study is interesting, but there are still some improvements to be made before it can be accepted for publication.

  1. Abstract, although the focus of this paper is on sintering kinetics, the results of microstructure and mechanical properties also account for a large part of the author's results. Please provide a concise summary of the findings to help readers quickly grasp the author's research.

Authors: We agree with this comment. The abstract has been revised.

  1. The introduction section is written well. The technological advantage of SPS in reducing sintering temperature is reflected in many materials. Below are a few references to help with this process. [1] “Densification mechanism of Zr-based bulk metallic glass prepared by two-step spark plasma sintering”, Journal of alloys and compounds, 2020, 850:156724. 10.1016/j.jallcom.2020.156724 [2] “Enhancing strength-ductility synergy in an ex situ Zr-based metallic glass composite via nanocrystal formation within high-entropy alloy particles”, Material and design, 2021, 210:110108. 10.1016/j.matdes.2021.110108

Authors: We agree with this comment. These references have been added to the Introduction.

  1. Figure 1, the XRD pattern for the powder sample TiO2 is incomplete. Please check.

Authors: We do not agree with this comment. XRD pattern for the TiO2 powder sample does not contain any reflex below 2θ< 20°

  1. Figure 2 is not clear, especially Figure 2d. Please provide clearer pictures of the powder morphology.

Authors: We agree with this comment. The resolution of our SEM equipment does not allow us to obtain higher quality images. Powders are accumulating an electrical charge during SEM, which also makes it difficult to obtain good quality images. Figure 2 has been excluded from the article because it does not carry any significant information.

  1. Line 160, The authors mentioned that the increase in the heating rate led to the shift of the curves towards high temperature by 150-200℃. The authors should specify the range of relative density corresponding to this offset temperature value; and it should be clearly indicated in the Figure 3. Why did the authors attribute this shift to the "imperfection" of the control system. Is there any literature or experimental data to support the author's opinion. The author's description of the delay of response in the hydraulic system is confusing. Please discuss it more detailly.

Authors:  We agree with this comment. Our opinion is based on extensive experience with a particular SPS equipment. We are currently preparing a separate article on this issue for publication. However, this problem is beyond the scope of this article. This item has been excluded from the article.

  1. Line 194, the effect of additive particles on the reduction of relative density is considerable. Please add some discussions about additive particles.

Authors:  We agree with this comment. In accordance with [Ashby M.F.; Verrall R.A. Diffusion-Accommodated Flow And Superplasticity. Acta Metallurgica 1973, 21, pp. 149-163] the second phase particles inhibit neck growing at the initial sintering stage and suppress the densification at the following sintering stages. This reference has been added.

  1. Figure 10, the angle range should be consistent with Figure 1. There is a spelling error in the title of Figure 10, please check

Authors:  We agree with this comment. The spelling error has been fixed.

The revisions in the paper are highlighted in yellow.

Best regards,

Maksim S. BOLDIN (Corresponding author)

PhD, Researcher,

Lobachevsky State University of Nizhny Novgorod

[email protected]

23 Gagarina ave., Nizhniy Novgorod, 603950, Russian Federation

Reviewer 3 Report

Thank you for submitting your paper. The work done here draws attention to a significant subject in alumina spark plasma sintering. I have found the paper to be interesting. However, several issues need to be addressed properly before the paper is being considered for publication. My comments including major and minor concerns are given below:

  1. Please consider reviewing the abstract and highlight the novelty, major findings, and conclusions. I suggest reorganizing the abstract, highlighting the novelties introduced. The abstract should contain answers to the following questions:
  2. What problem was studied and why is it important?
  3. What methods were used?
  4. What conclusions can be drawn from the results? (Please provide specific results and not generic ones).
  5. The abstract must be improved. It should be expanded. Please use numbers or % terms to clearly shows us the results in your experimental work. Please expand the abstract.
  6. Please consider reporting on studies related to your work from mdpi journals.
  7. The introduction must be expanded, please consider improving the introduction, provide more in-depth critical review about past studies similar to your work, mention what they did and what were their main findings then highlight how does your current study brings new difference to the field.
  8. The authors write so many small paragraphs of 1-3 lines, this is not acceptable. It distorts the readability of the manuscript. The authors must combine all small paragraphs into larger ones. Please check for this issue everywhere in the manuscript.
  9. Combine Lines 84-96 into one larger paragraph.
  10. Combine Lines 97-112 into one larger paragraph.
  11. Combine Lines 113-130 into one larger paragraph.
  12. Change 3. Results  to 3. Results and Discussion
  13. Line 132 please improve this title “3.1. Powder research” maybe say XRD analysis or XRD analysis of alumina powder.
  14. Figure 2 add some text and arrows to indicate to the readers regions of interest on those SEM graphs.
  15. Title “3.2 Sintering of ceramics” must be changed, it does not reflect what was done in this sub-section.
  16. Lines 154-155 “have a monotonic character up to the temperature…” why, please discus this further and support with references, also was this observed in past studies similar to this work or not?
  17. Lines 160-161 we can already see what you are telling us from the graphs so what is interesting about this observation?
  18. Line 161 remove “In our opinion”.
  19. Line 162 “Dr. Sinter model SPS-625” is using the word Dr here is correct?
  20. Lines 162-165 I am not convinced with this explanation, this is machine issue or way of operating? What about other factors? If the reason is the machine performance, then did the authors try to use another machine and compare the differences?
  21. Lines 160-177 combine into one larger paragraph.
  22. Line 173 “An increase in the heating rate leads to an increase in Smax for all powders.” Why? The authors should discuss this further and support with references.
  23. Figures 6-9 either move to an appendix or combine all in one larger figure as there are so many of them one after the other without much discussion on them. Also add some arrows and text to tell the readers what they are looking at in those SEM graphs.
  24. Line 205 “does not affect the size of the grain.” Why? What is the onset of temperature after which the grain size is not affected further and why this happens?
  25. Lines 216-226 combine into one larger paragraph.
  26. Figure 10 is not clear, please improve quality.
  27. Lines 228-243 combine into one larger paragraph.
  28. Lines 246-274 combine into one larger paragraph.
  29. The authors should add a list of nomenclature at the end of the manuscript which contains all symbols, Greek letters, abbreviations ..etc.
  30. Lines 300-314 combine into one larger paragraph.
  31. Line 305 “independent on the” change to “independent from”
  32. It is strongly recommended that the authors combine the results and discussion sections together for ease of readability of the article findings.
  33. Section 4.3. Second stage analysis can be moved to an appendix, only keep important information like table 4.
  34.  Lines 375-390 combine into one larger paragraph.
  35. Some of results are merely described and is limited to comparing the experimental observation and describing results. The authors are encouraged to include a more detailed results and discussion section and critically discuss the observations from this investigation with existing literature.
  36. Conclusion can be expanded or perhaps consider using bullet points (1-2 bullet points) from each of the subsections.

Author Response

Response to the Reviewer's comments

Dear Reviewer!

Thank you for your thorough analysis of our article. We agree with most of your comments and, in accordance with them, we have conducted a revision of our article. Below are our responses to the Reviewer's comments.

Reviewer #3

Thank you for submitting your paper. The work done here draws attention to a significant subject in alumina spark plasma sintering. I have found the paper to be interesting. However, several issues need to be addressed properly before the paper is being considered for publication. My comments including major and minor concerns are given below:

  1. Please consider reviewing the abstract and highlight the novelty, major findings, and conclusions. I suggest reorganizing the abstract, highlighting the novelties introduced. The abstract should contain answers to the following questions:
  2. What problem was studied and why is it important?
  3. What methods were used?
  4. What conclusions can be drawn from the results? (Please provide specific results and not generic ones).
  5. The abstract must be improved. It should be expanded. Please use numbers or % terms to clearly shows us the results in your experimental work. Please expand the abstract.

Authors:  We agree with this comment. The abstract has been revised.

  1. Please consider reporting on studies related to your work from mdpi journals.

Authors:  We agree with this comment. The following references have been added.

Tokita, M. Progress of Spark Plasma Sintering (SPS) Method, Systems, Ceramics Applications and Industrialization. Ceramics 2021, 4, 160-198. https://doi.org/10.3390/ceramics4020014

Lantcev, E.; Nokhrin, A.; Malekhonova, N.; Boldin, M.; Chuvil’deev, V.; Blagoveshchenskiy, Y.; Isaeva, N.; Andreev, P.; Smetanina, K.; Murashov, A. A Study of the Impact of Graphite on the Kinetics of SPS in Nano- and Submicron WC-10%Co Powder Compositions. Ceramics 2021, 4, 331-363. https://doi.org/10.3390/ceramics4020025

  1. The introduction must be expanded, please consider improving the introduction, provide more in-depth critical review about past studies similar to your work, mention what they did and what were their main findings then highlight how does your current study brings new difference to the field

Authors:  We agree with this comment. The following changes have been added.

Typically, sintering models consider a medium heating rate of powders with a grain size of up to 1 µm [Rahaman M.N. Ceramic Processing and Sintering Second edition. Publisher: Marcel Dekker Inc 2003, pp. 876]. These models assume that the dominant mechanism of shrinkage in the final stage of sintering is a creep. However, the joint use of high heating rates and submicron powders can lead to a change in densification mechanisms. Thus [Stuer M.; Carry C.P.; Bowen P.; Zhao Z. Comparison of apparent activation energies for densification of alumina powders by pulsed electric current sintering (spark plasma sintering) and conventional sintering—toward applications for transparent polycrystalline alumina, J. Mater. Res. 2017, pp. 3309-3318] report about viscous flow mechanism during spark plasma sintering of alumina, but do not reveal its nature. In the range of temperatures and heating rates that are used in this study, the densification process can be described from the point of view of grain boundary sliding. It should be noted that a similar result was found for SPS nanopowders of zirconia [Bernard-Granger G.; Guizard C. Spark plasma sintering of a commercially available granulated zirconia powder: I. Sintering path and hypotheses about the mechanism(s) controlling densification, Acta Materials 55 3493, 2007, pp. 3493-3504].

The aim of this work is to analyze the densification behavior of fine alumina powder from a grain boundary sliding mechanism point of view. The authors also try to study the stages of spark plasma sintering of alumina powders, as well as to study the effect of par-ticles of the second phase on the sintering kinetics at different stages of sintering.

  1. The authors write so many small paragraphs of 1-3 lines, this is not acceptable. It distorts the readability of the manuscript. The authors must combine all small paragraphs into larger ones. Please check for this issue everywhere in the manuscript.
  2. Combine Lines 84-96 into one larger paragraph.
  3. Combine Lines 97-112 into one larger paragraph.
  4. Combine Lines 113-130 into one larger paragraph.

Authors:  We agree with this comment. The it has been fixed.

  1. Change 3. Results  to 3. Results and Discussion

Authors:  We agree with this comment. The title has been fixed.

  1. Line 132 please improve this title “3.1. Powder research” maybe say XRD analysis or XRD analysis of alumina powder.

Authors:  We agree with this comment. The title has been fixed.

  1. Figure 2 add some text and arrows to indicate to the readers regions of interest on those SEM graphs.

Authors:  We agree with this comment. The resolution of our SEM equipment does not allow us to obtain higher quality images. Figure 2 has been excluded from the article because it does not carry any significant information.

  1. Title “3.2 Sintering of ceramics” must be changed, it does not reflect what was done in this sub-section.

Authors:  We agree with this comment. The subtitle has been fixed.

  1. Lines 154-155 “have a monotonic character up to the temperature…” why, please discus this further and support with references, also was this observed in past studies similar to this work or not?

Authors:  We agree with this comment. Such type of densification cure is common for solid phase sintering of polycrystalline materials. The reverence has been added.

  1. Lines 160-161 we can already see what you are telling us from the graphs so what is interesting about this observation?
  2. Line 161 remove “In our opinion”.
  3. Line 162 “Dr. Sinter model SPS-625” is using the word Dr here is correct?
  4. Lines 162-165 I am not convinced with this explanation, this is machine issue or way of operating? What about other factors? If the reason is the machine performance, then did the authors try to use another machine and compare the differences?

Authors:  We agree with this comment. Our opinion is based on extensive experience with a particular SPS equipment. We are currently preparing a separate article on this issue for publication. However, this problem is beyond the scope of this article. Lines 160-165 has been excluded from the article.

  1. Lines 160-177 combine into one larger paragraph.

Authors:  We agree with this comment. Lines 160-165 has been excluded from the article.

Lines 165-177 has been combined.

  1. Line 173 “An increase in the heating rate leads to an increase in Smax for all powders.” Why? The authors should discuss this further and support with references.

Authors:  We agree with this comment. The following comments have been added.

This result correlate with [Olevsky E.A.; Kandukuri S.; Froyen. L.  Consolidation enhancement in spark-plasma sintering: Impact of high heating rates. Journal of Applied Physics 102 2007, pp. 1-13]. When using high heating rates, the isolated powder particles are placed under the action of the high temperatures, at this moment the necks between the particles are formed and the saved surface curvature of the particles provides an additional driving force for sintering and Smax increases.

  1. Figures 6-9 either move to an appendix or combine all in one larger figure as there are so many of them one after the other without much discussion on them. Also add some arrows and text to tell the readers what they are looking at in those SEM graphs.

Authors:  We agree with this comment. Figures 6-9 have been moved to the Appendix A. The arrows have been added.

  1. Line 205 “does not affect the size of the grain.” Why? What is the onset of temperature after which the grain size is not affected further and why this happens?

Authors:  We agree with this comment. The following comment has been added.

This is because the scale of grain change with increasing heating rate from 350 С/min to 700 С/min is less than the measurement error.

  1. Lines 216-226 combine into one larger paragraph.
  2. Figure 10 is not clear, please improve quality.
  3. Lines 228-243 combine into one larger paragraph
  4. Lines 246-274 combine into one larger paragraph.

Authors:  We agree with this comment. The lines have been combined.

  1. The authors should add a list of nomenclature at the end of the manuscript which contains all symbols, Greek letters, abbreviations ..etc.

Authors:  We agree with this comment. The list of nomenclature has been added to the Appendix

  1. Lines 300-314 combine into one larger paragraph.

Authors:  We agree with this comment. The lines have been combined.

  1. Line 305 “independent on the” change to “independent from”

Authors:  We agree with this comment. It has been fixed.

  1. It is strongly recommended that the authors combine the results and discussion sections together for ease of readability of the article findings.

Authors:  We agree with this comment. Section 3 and Section 4 have been combined.

  1. Section 4.3. Second stage analysis can be moved to an appendix, only keep important information like table 4.

Authors:  We do not agree with this comment. This section is necessary.

  1. Lines 375-390 combine into one larger paragraph.

Authors:  We agree with this comment. The lines have been combined.

  1. Some of results are merely described and is limited to comparing the experimental observation and describing results. The authors are encouraged to include a more detailed results and discussion section and critically discuss the observations from this investigation with existing literature.

Authors:  We agree with this comment. The following comment has been added.

The result obtained is correlate with the results of [Stuer] where the viscous flow was called as a major mechanism of spark plasma sintering of alumina. but authors [Stuer] do not reveal its nature. Viscous flow should have led to obtaining the elongated grains which were not observed. Another kind of such "plastic deformation" without elongation of grains can be caused by grain boundary sliding during sintering. It should be noted that a similar result was found for SPS nanopowders of zirconia [Bernard].

Since grain growth occurs at the third stage of sintering, the densification behavior at this stage will depend on the grain growth kinetics (the ratio of the grain boundary migration rate and the pore migration rate) [Rahaman]. For this reason, the analysis of the third stage of sintering is excluded from consideration in this article.

  1. Conclusion can be expanded or perhaps consider using bullet points (1-2 bullet points) from each of the subsections.

Authors:  We agree with this comment. Conclusion has been expanded.

The revisions in the paper are highlighted in yellow.

Best regards,

Maksim S. BOLDIN (Corresponding author)

PhD, Researcher,

Lobachevsky State University of Nizhny Novgorod

[email protected]

23 Gagarina ave., Nizhniy Novgorod, 603950, Russian Federation

Round 2

Reviewer 1 Report

Thank you for providing revised manuscript. I am satisfied with all the comments.

Author Response

Dear Reviewer,
Thank you.

Reviewer 2 Report

Publish it.

Author Response

Dear Reviewer,

Thank you!

Reviewer 3 Report

Dear authors, some minor modifications, after that the paper can be accepted. Congratulations:

  1. In introduction, the authors add bulk citations in many locations, lines 40/47/53/63. This is not acceptable and should be reduced (1-2 references) max especially if it just a generic information. Otherwise, the authors need to give full credit for those many cited articles.
  2. Figure 6 please add some arrows and text on those images to explain to the readers what they are looking at into them. Same goes for Figure 7.
  3. Combine lines 38-47 in one larger paragraph.

  4. Combine lines 48-59 in one larger paragraph. 

Author Response

Dear Reviewer!

Thank you for your thorough analysis of our article. We agree with most of your comments and, in accordance with them, we have conducted a revision of our article. 
